# Ensemble-Based Virtual Screening Led to the Discovery of Novel Lead Molecules as Potential NMBAs

**DOI:** 10.3390/molecules29091955

**Published:** 2024-04-24

**Authors:** Yi Zhang, Gonghui Ge, Xiangyang Xu, Jinhui Wu

**Affiliations:** 1School of Medicine, Nanjing University, Nanjing 210093, China; cruckzhang0304@163.com; 2Jiangsu Key Laboratory of Central Nervous System Drug Research and Development, Jiangsu Nhwa Pharmaceutical Co., Ltd., Xuzhou 221116, China; 3School of Pharmacy, China Medical University, Shenyang 110122, China

**Keywords:** NMBAs, pharmacophore model, molecular docking, molecular dynamics, virtual screening

## Abstract

Neuromuscular blocking agents (NMBAs) are routinely used during anesthesia to relax skeletal muscle. Nicotinic acetylcholine receptors (nAChRs) are ligand-gated ion channels; NMBAs can induce muscle paralysis by preventing the neurotransmitter acetylcholine (ACh) from binding to nAChRs situated on the postsynaptic membranes. Despite widespread efforts, it is still a great challenge to find new NMBAs since the introduction of cisatracurium in 1995. In this work, an effective ensemble-based virtual screening method, including molecular property filters, 3D pharmacophore model, and molecular docking, was applied to discover potential NMBAs from the ZINC15 database. The results showed that screened hit compounds had better docking scores than the reference compound *d*-tubocurarine. In order to further investigate the binding modes between the hit compounds and nAChRs at simulated physiological conditions, the molecular dynamics simulation was performed. Deep analysis of the simulation results revealed that ZINC257459695 can stably bind to nAChRs’ active sites and interact with the key residue Asp165. The binding free energies were also calculated for the obtained hits using the MM/GBSA method. In silico ADMET calculations were performed to assess the pharmacokinetic properties of hit compounds in the human body. Overall, the identified ZINC257459695 may be a promising lead compound for developing new NMBAs as an adjunct to general anesthesia, necessitating further investigations.

## 1. Introduction

Neuromuscular blocking agents (NMBAs), commonly referred to as muscle relaxants, are frequently used to facilitate tracheal intubation and provide skeletal muscle relaxation during surgery or mechanical ventilation [1]. There are two primary categories of NMBAs, defined according to their blocking mechanisms: depolarizing and nondepolarizing agents [2]. Succinylcholine is the only available depolarizing NMBAs still in clinical use. Due to its rapid onset and short duration, the utility of succinylcholine is limited by mechanism-related side effects, such as myalgia, hyperkalemia, and malignant hyperthermia [3,4]. The majority of currently used NMBAs are nondepolarizing blockers (Figure 1), classified structurally into benzylisoquinolines (e.g., cisatracurium and mivacurium) and aminosteroids (e.g., rocuronium, vecuronium, pipecuronium, and pancuronium). Nondepolarizing blockers are characterized by two quaternary ammonium groups, and the distance between the two protonated nitrogen atoms is approximately 14 Å [5], corresponding to a 10-atom separation (“14-Å rule” or “10-atom rule”) [6]. The nondepolarizing NMBAs inhibit nicotinic acetylcholine receptors (nAChRs) located postsynaptically on skeletal muscle membranes, thereby inducing skeletal muscle relaxation [5].

Muscular nAChRs are ligand-gated pentameric ion channels physiologically activated by acetylcholine (ACh), and the activation of nAChRs initiates the electrical signal that triggers action potentials, leading to muscle contraction [7]. As shown in Figure 2A (PDB ID: 7SMS), muscular nAChRs structurally consist of five protein subunits organized around a central pore: with two alpha (2α) subunits, one beta (β) subunit, and one delta (δ) subunit, accompanied by either an epsilon (ε) or a gamma (γ) subunit, depending on the developmental stage of the muscle [8,9]. The adult muscle receptors have an epsilon subunit, whereas in infants the gamma type is present [10]. Each muscular nAChR has two binding pockets for ACh in the extracellular domain, which are located at the interfaces of the α-δ and α-ε(γ) subunits [11]. In these pockets, electron-rich amino acid residues (e.g., tyrosine, tryptophan) can interact electrostatically with the positively charged ammonium group of ACh [12,13]. The nondepolarizing NMBAs act as competitive antagonists and impede the interaction between ACh and nAChRs through occupying the ACh binding sites, thereby preventing the opening of the channel (Figure 2B). In fact, positive charges at the quaternary ammonium sites of NMBAs mimic the quaternized nitrogen atom of ACh, which is the structural reason for the attraction of these compounds to muscular nAChRs [14,15].

Currently, muscle relaxation remains a mainstay of modern anesthesia and intensive care [16,17]. NMBAs are used to ease tracheal intubation and to decrease the doses of the general anesthetics [18,19]. Other uses include in acute respiratory distress syndrome (ARDS) [20], elevated intracranial pressure (ICP) [21], and therapeutic hypothermia (TTM) [22]. To meet different pharmacological needs, a variety of structurally diverse compounds have been reported as potential NMBAs [14,23,24,25]. However, the progress in NMBAs development has been impeded since the discovery of cisatracurium over 25 years ago. High-throughput screening techniques, especially computer-aided virtual screening, have been widely employed for the discovery of lead molecules with new scaffolds of specified targets from large chemical databases [26]. These computer-assisted design methods can be classified as ligand- and structure-based virtual screening approaches [27,28]. The ligand-based virtual screening (LBVS) methods, including pharmacophore modeling and qualitative structure–activity relationships (QSAR), focus on comparative molecular similarity analysis of compounds with unknown and known activity [29,30]. The structure-based virtual screening (SBVS) method, such as molecular docking, is an effective tool to discover putative targets for a particular ligand [31]. Up to now, hundreds of compounds with neuromuscular blocking activities have been reported by many research groups (Appendix A). Furthermore, high-resolution cryo-EM structures of muscle-type nicotinic receptor with *d*-tubocurarine have recently been solved (PDB ID: 7SMS) [32]. Therefore, in this study, molecular property filtering and 3D pharmacophore modelling were applied as ligand-based screening techniques to identify compounds with similar characteristics to known neuromuscular blockers. Subsequently, structure-based docking studies were conducted to assess their binding affinities and poses. As shown in Figure 3, the three virtual screening methods were combined to discover potential NMBAs from ZINC15 database. Moreover, molecular dynamics simulation was applied to investigate the stability of the complexes and the mechanism of interaction between nAChRs and hit compounds at physiological conditions.

## 2. Results

### 2.1. Molecular Property Filters

Muscle relaxants are mainly quaternary ammonium compounds with a large molecular size, making them unique among the drugs associated with disease [33]. For this reason, classical screening strategies, such as the Lipinski’s rule of five, may not be appropriate for the discovery of promising lead molecules. Therefore, a dataset of 294 reported compounds with neuromuscular blocking activities was initially compiled from the earlier literature (Appendix A). Then, three key properties (molecular weight, charge, and logP) of compounds in the NMBAs database were calculated using the ChemDraw software (Version 18.2). The 997 million compounds in the ZINC15 database were filtered as preliminary screening based on the 3 properties. As we can see in Figure 4A, compounds in the database have large molecular weights, with only a few less than 400 Da, so a filter was applied to exclude compounds with a molecular weight below this threshold. Given that NMBAs are primarily composed of bis-quaternary ammonium salts, as shown in Figure 4B, 93.9% of compounds in the database are double-charged cationic molecules. Consequently, all double positively charged compounds were selected to ensure that the molecules had desirable physicochemical properties as drug molecules. Figure 4C shows that the majority of compounds with calculated logP values fall within the range of −4 to 4; a third filter was, therefore, employed to exclude the compounds with logP values outside of this window. The filtration resulted in obtaining nearly 562 thousand compounds from the ZINC15 database in SDF format. In addition, the compounds were minimized using MOE software (Version 2019.01) and then selected for virtual screening through the pharmacophore model.

### 2.2. Pharmacophore Modelling

A pharmacophore is a representation of steric and electronic features required for interaction with a macromolecular target, which results in a pharmacological response [34]. We developed a ligand-based pharmacophore model in MOE based on the six marketed nondepolarizing NMBAs mentioned in Figure 1. The selection of these compounds to build the pharmacophore model was based on their high potency and structural diversity. The 3D pharmacophore was generated by the flexible alignment of all six structures, and the 3D features that they shared were identified through the pharmacophore consensus module [35]. The common 3D pharmacophore model is composed of four hydrophobic groups (F1, F2, F3, F4), one aromatic ring (F5), and two cations (F6, F7). In addition, previous study has indicated that the distance of the two quaternized N atoms falling in between 11 and 14 Å is desirable for maximizing bioactivity [6,12]. As shown in Figure 5, the distance between F6 and F7 falls between 11.7 and 15.3 Å, which was in compliance with the “14-Å rule”. The model offered a molecular framework for the virtual screening of databases utilizing suitable pharmacophore features (Figure 5). A total of 562 thousand chemical compounds selected from the ZINC15 database were screened through the generated 3D pharmacophore model. As a result of pharmacophore-based virtual screening, 2476 hit molecules were identified to fit the pharmacophore features.

### 2.3. Molecular Docking

Molecular docking provides important data in rational drug design, which can predict the binding affinities, spatial orientation, and predominant binding modes of the small molecule drug candidates to the active site of target proteins [36]. In this investigation, we employed structure-based virtual screening approach that involves a docking simulation of 2476 molecules identified through the pharmacophore model into the active pocket of nAChRs (PDB ID: 7SMS) [32]. Muscular nAChRs contain two binding sites in the extracellular domain (ECD), which are located at the α-γ and α-δ (infant) or α-δ and α-ε (adult) subunit interfaces, respectively [8]. As shown in Figure 6, the transmembrane domain (TMD) of each subunit comprises four helices, M1-M4, with M2 lining the ion channel and M4 being most peripheral [32]. The intracellular domain (ICD) of each subunit is formed by a partially ordered loop between the M3 and M4 helices: an amphipathic MX helix following M3, and a long helix called MA that leads into and is continuous with M4. The MA helices form a bundle at their N-termini and frame lateral portals for ion flux [37]. We performed molecular docking simulations of the compounds binding to the nAChRs α-δ site because it is present in both these nAChR subtypes [38]. The computationally derived potential hits, along with the binding energy values, were shown in Figure 6. As lower binding energy corresponds to a higher binding affinity, four compounds, namely ZINC257357801, ZINC257459695, ZINC8926303, and ZINC1293069436, emerged as the best compounds with better scoring energies (−8.60 kcal/mol, −8.52 kcal/mol, −8.36 kcal/mol, and −8.25 kcal/mol, respectively) than the standard compound d-tubocurarine (−8.14 kcal/mol).

The four compounds with desired properties were selected as potential NMBAs by the ensemble-based virtual screening strategy. The molecular interactions between the hits and nAChRs produced by MOE are displayed in Figure 7. Through the analysis of the cryo-EM structure, we found that the quaternized NH group of d-tubocurarine can form hydrogen bonds with the key residue Asp165 (Appendix A), which played important roles in binding. As shown in Figure 7A, residue Asp165 formed a hydrogen bond with the dimethylamino group of ZINC257357801, and residues of Cys192 and Trp149 formed hydrogen bonds with the hydrogen atom on the hydroxyl group in the middle and right side, respectively. The residue Tyr198 formed a Pi-H bond with the ethyl group, and residue Trp57 formed a Pi-cation with the quaternized nitrogen atom of the ring. For ZINC257459695, as shown in Figure 7B, residue Asp165 formed a hydrogen bond with the hydrogen atom of the pyrrolidine ring, residue Cys192 formed a hydrogen bond with the oxygen atom on the carbonyl group, and residue Asp180 formed a hydrogen bond with the methyl ester at the right end. For ZINC8926303, residues of Asp165 and Ile148 formed hydrogen bonds with hydrogen atoms on two terminal quaternary ammonium groups, residue Tyr190 formed a Pi-H bond with the piperazine ring, and residue Tyr198 formed a Pi-H bond with the ethyl group in the right side. For ZINC1293069436, residue Asp59 formed a hydrogen bond with the hydrogen atom on the methylene group, residue Tyr198 formed a hydrophobic interaction with the benzene ring, and residue Met163 formed a Pi-H bond with the pyrazole ring. Obviously, it was evident from Figure 7 that ZINC257357801, ZINC257459695, and ZINC8926303 could interact with the key residue Asp165 in the active site of nAChRs. Therefore, the stability and molecular interaction pattern of these three hits and d-tubocurarine at simulated physiological condition were further probed by applying molecular dynamic (MD) simulation.

### 2.4. Molecular Dynamic Simulation

In this section, we conducted a molecular dynamics simulation to investigate the flexibility and solvation effect of biomolecular systems. To this end, we ran 25,000 ps molecular dynamics production runs on the 3 hit candidates (ZINC257357801, ZINC257459695, and ZINC8926303) and the reference compound (d-tubocurarine). The MD trajectories were analyzed by the flowing parameters: root mean square deviation (RMSD), root mean square fluctuation (RMSF), radius of gyration (Rg), solvent accessible surface area (SASA), hydrogen bonds (H-bond), dynamic cross-correlation matrix (DCCM), and binding pattern analysis.

#### 2.4.1. Structural Deviation and Compactness Analysis

The structural deviation and stability were measured by the RMSD and SASA, and the fluctuation was assessed by RMSF, and the compactness was measured by Rg. From the observation of the RMSD graph presented in Figure 8A, all the protein–ligand entities showed steady and stable behavior throughout the simulation time; the RMSD fluctuated around by 0.15–0.35 nm. However, in the same simulation time, the ZINC257459695 graph line showed much stable behavior as compared to other complexes, which remained steady and stable at an RMSD value of around 0.18 nm. The RMSD graph lines of ZINC257357801, ZINC8926303, and *d*-tubocurarine displayed an increasing trend, with RMSD values ranging from 0 to 0.34 nm from 0 to 5000 ps. The overall RMSD graphs results showed stable behavior in the backbone of all docked complexes.

The root mean square fluctuation calculates the fluctuation in the individual amino acid residues throughout the simulation process in the presence of different ligand molecules. As represented in Figure 8B, all the promising lead molecules show a very similar RMSF pattern with the reference compound. Some residues in the MA and M4 domains (chain A), and MA domain (chain B) showed greater flexibility, and RMSF fluctuated between 0.11 and 0.76 nm. The α-helixes of MA and M4 are located around the frame lateral portals for ion flux, which may be the reason for the flexibility.

The Rg parameter was used to evaluate structural compactness of a protein–ligand in a biological system. A stable folded structure is described by a relatively constant Rg value, whereas an unfolded structure will cause the Rg value to fluctuate with time. As shown in Figure 8C, the values of Rg for the four complexes fluctuated between 4.09 and 4.17 nm during the simulation. The calculated average Rg values for ZINC257357801, ZINC257459695, ZINC8926303, and *d*-tubocurarine are 4.15 nm, 4.11 nm, 4.17 nm, and 4.13 nm, respectively. With lower Rg values compared to all other compounds, ZINC257459695-nAChRs may be regarded as the most compact biomolecular system.

The solvent accessible surface area of protein has always been considered as a decisive factor in protein folding and stability studies. As we can see in Figure 8D, the SASA for the four systems showed little variation from 390 to 425 nm^2^, which suggested that the four systems were relatively stable. The binding of ZINC257459695, compared to other compounds, resulted in reduced SASA values, as the surface of the protein becomes unexposed to the solvent after ligand binding.

#### 2.4.2. Hydrogen Bond Analysis

The stability of the protein–ligand complex is facilitated by the formation of hydrogen bonds between the receptor and ligand. Therefore, the total number of hydrogen bonds were investigated in the complexes after the 25,000 ps simulation time. As exhibited in Figure 9A,D, for the ZINC257357801 and *d*-tubocurarine complexes, two to three hydrogen bonds were identified. ZINC8926303, on the other hand, was shown to form three to four hydrogen bonds. Interestingly, for the ZINC257459695–nAChRs system, a maximum of five hydrogen bonds were observed, which could form more hydrogen bonds than that of the reference compound *d*-tubocurarine during the entire simulation period. Obviously, the three hits could form more hydrogen bonds than the reference compound. Furthermore, through the above-detailed H-bond analysis, we can conclude that the compound ZINC257459695 was bound to the nAChRs protein more effectively and tightly when compared to the other three compounds. The results of the H-bond analysis were consistent with the earlier analysis of RMSD and SASA metrics.

#### 2.4.3. Dynamic Cross-Correlation Map Analysis

The analysis of DCCM was to check the correlated motion of structural domains to achieve a stable conformation of the receptor following the binding of ligands. Highly positive sections (cyan) indicate strongly positive correlated movement of residues in the same direction, while the negative regions (pink) represent strongly anti-correlated motions. As illustrated in Figure 10, the binding of ZINC257357801, ZINC257459695, ZINC8926303, and *d*-tubocurarine generated obvious influences on the internal dynamic behavior of nAChRs. For the ZINC257459695 complex, both chain A and B produced strongly positive correlated motions independently (Figure 10B). This phenomenon suggested that the binding of ZINC257459695 may have resulted in conformational changes in the protein. The ZINC257357801 complex has maximum residues in a positive correlation, while ZINC8926303 complex residues have a slightly weaker but still positive correlation (Figure 10A,C). From the results, it is concluded that the presence of ZINC257357801 and ZINC8926303 induced significant correlated motions in protein, whereas slightly anti-correlated motions are observed. The *d*-tubocurarine complex has the most residues in a negative correlation, but the overall correlation between the binding site and the amino acids has increased (Figure 10D).

#### 2.4.4. The Binding Modes Refined through the MD Simulations

The Gibbs free energy landscape for the four systems was displayed in Figure 11A–D. A very weak or unstable receptor–ligand interaction can result in many minimal energy clusters, whereas a strong and stable interaction can generate one conformation cluster in the potential energy map [39]. As depicted in Figure 11, only a single energy minima was found in the case of the ZINC257357801, ZINC257459695, and ZINC8926303 complexes, whereas there are two energy minima found for the *d*-tubocurarine complex. In addition, the narrow and shallow energy basin denotes limited structural conformation stability. It can be observed from Figure 11 that the binding of the nAChRs protein with ZINC257357801 and ZINC257459695 has resulted in a noticeable single narrow energy minima basin related to its conformational state, suggesting a stable and strong receptor–ligand conformation. Although the *d*-tubocurarine complex has two energy minima, they are completely separated from one another by an energy barrier making it less stable. This is also in agreement with earlier RMSD, H-bond, and DCCM analysis results.

Based on the calculation of the Gibbs free energy, the most stable conformations of each system were extracted from the lowest energy field (dark blue) to explore the key residues and interactions between these compounds and nAChRs. For the ZINC257357801 system, as shown in Figure 12A, residue Asp165 formed two different hydrogen bonds with the dimethylamino group. ZINC257357801 and Trp57 formed a Pi-H interaction, and residue Trp149 formed a hydrogen bond with the hydroxyl group at the right end. As exhibited in Figure 12B, based on the most energetically favorable binding mode, residues of Asp165 and Cys193 formed two hydrogen bonds with ZINC257459695, while residues of Tyr198 and Trp57 were observed to form hydrophobic interactions. For the ZINC8926303 system, as shown in Figure 12C, ZINC8926303 formed hydrophobic interactions with residues of Trp149, Tyr151, and Cys193. Notably, the hydrogen bond between the terminal quaternary ammonium group and residue Asp165 disappeared, which could be the reason for the decreased stability after the simulation. For the *d*-tubocurarine system, as displayed in Figure 12D, the quaternized NH group acted as a hydrogen bond donor with residue Asp165, and Tyr190 interacted with *d*-tubocurarine as a hydrophobic interaction.

#### 2.4.5. Binding Free Energy Calculation by MM/GBSA Method

To better assess the reliability of the different nAChRs–ligand complexes, the corresponding protein–ligand binding free energies were evaluated from the MD coordinates extracted from the last 10 ns of simulation. The molecular mechanics generalized Born surface area (MM-GBSA) method was used for the calculation. The binding free energy Δ*G*_bind_ was comprised of the Van der Waals interaction (Δ*E*_vdw_), electrostatic interaction (Δ*E*_ele_), polar solubility energy (Δ*G*_ps_), and non-polar solubility energy (Δ*G*_nps_). Among these interactions, Δ*E*_vdw_, Δ*E*_ele_, and Δ*G*_nps_ were considered to be beneficial for the Δ*G*_bind_, but Δ*G*_ps_ was considered to be unfavorable to the Δ*G*_bind_. The analysis of the MM-GBSA results showed that the electrostatic force of interaction is majorly contributing in the protein–ligand binding compared to the Van der Waals force of interaction. As shown in Table 1, ZINC257459695 had a maximum affinity with nAChRs, followed by ZINC257357801 and ZINC8926303, showing −50.40 ± 3.61, −40.52 ± 5.49, and −31.01 ± 6.22 kcal/mol, respectively. The estimated total binding energy suggested that the ZINC257459695 has the most energetically favored binding mode as compared to the reference, *d*-tubocurarine. However, compared with the cisatracurium and rocuronium, it can be seen that the Δ*E*_vdw_ and Δ*G*_nps_ of ZINC257459695 were much lower, indicating that the two marketed NMBAs possess superior binding energy. In addition, the binding free energy of the ZINC8926303 is the weakest. The Δ*E*_ele_ of ZINC8926303 complex was the lowest for all three hit compounds, mainly because the Δ*G*_bind_ of the ZINC8926303 was weaker than that of the standard *d*-tubocurarine.

### 2.5. In Silico Pharmacokinetic Profile (ADMET)

In the pursuit of efficient drug discovery, data on absorption, distribution, metabolism, excretion, and toxicity (ADMET) are crucial for identifying and developing novel drug candidates [40,41]. Thus, the pharmacokinetic profile of the top three hits, along with *d*-tubocurarine, cisatracurium, and rocuronium, were assessed using the PreADMET method [42] and summarized in Table 2. The aqueous solubility of a drug is a vital factor that can significantly affect its bioavailability, and the three hit compounds exhibited moderate to high water solubility levels. Blood–brain barrier penetration was used to evaluate the distribution of the compounds, and the BBB values for ZINC257357801, ZINC257459695, and ZINC8926303 were 0.04, 0.11, and 0.05, respectively. During the metabolic phase, the ZINC257357801, ZINC8926303, and rocuronium were discovered to be inhibitors of cytochrome P450 2D6. In addition, the Ames test result of ZINC257459695 was negative, implying that it was probably unable to induce gene mutation. However, the usage of ZINC257459695 could be limited due to its blockage of hERG channels, warranting further optimization.

## 3. Materials and Methods

### 3.1. Ligand-Based Pharmacophore Generation

The alignments of six marketed NMBAs (i.e., cisatracurium, mivacurium, rocuronium, vecuronium, pipecuronium, and pancuronium) were generated using the Flexible Alignment module of the MOE (Molecular Operating Environment software, Version 2019.01) [43]. The 3D chemical structures of the NMBAs were built and energy-minimized using the MMFF94 force field [44]. Then, small molecules were aligned by maximizing the structural overlap of steric features while exploring alternative conformations with low ligand strain. Finally, a collection of alignments, along with a score for each alignment, was generated, and a final alignment with the lowest S score was obtained for the following pharmacophore generation.

The identification of common structural features among six aligned NMBAs was performed using a ligand-based pharmacophore approach. To generate a pharmacophore model with good quality, a set of pharmacophore features, namely hydrogen-bond donor (Don), hydrogen-bond acceptor (Acc), aromatic center (Aro), hydrophobic atom (HydA), anionic atom (Ani), and cationic atom (Cat) were mapped. The Pharmacophore Consensus module was used to generate suggested features for a pharmacophore query from the aligned structures, and a pharmacophore model containing seven common chemical features was developed. Based on the pharmacophore model generated, virtual screening was conducted using a Pharmacophore Search protocol in MOE through the EHT scheme [45]. A pharmacophore query consisting of identified pharmacophoric features was used to filter the database of molecular conformations.

### 3.2. Molecular Docking

The cryo-EM structures of muscle-type nicotinic receptor (PDB ID: 7SMS, Resolution 3.18 Å) [32] were obtained from the RCSB Protein Data Bank (PDB, https://www.rcsb.org/, accessed on 3 January 2024). As a part of preparing the target protein, the water molecules were removed from the protein and hydrogen atoms were added to optimize the structure by using energy minimization. For the docking parameters, we set the force field to Amber10 and used the triangle matcher placement algorithm, which returned thirty poses; we further used the rigid receptor refinement method, which returned five poses [32]. The GBVI/WSA dG method was applied to score the poses in both steps [46]. The scores were generated by accounting for the individual contributions of energy terms, including hydrogen bonds, electrostatics, and hydrophobicity. The scoring of ligands determines which ligand pose is the most energetically favorable and ranks the library of screened molecules to indicate which compounds are most likely to be active and suitable for further analysis. The selected complexes were illustrated using the PyMOL software (Version 2.5.0) [47].

### 3.3. Molecular Dynamics Simulation

All simulations were performed with the GROMACS 2019.6 package [48] (https://www.gromacs.org, accessed on 2 February 2024) and were carried out using the amber99sb-ildn force field at 298 K. General amber force field (GAFF) parameters were assigned to the ligands [49], whereas partial charges were calculated using the AM1-BCC method [50]. All ligand–protein complexes were placed in the cubic water-box and set to be 15 Å away from the box edge, using a TIP3P explicit solvent model [51]. Either Na^+^ or Cl^−^ ions were added as counterions for the neutralization of the systems. A 2 fs time step of integration was chosen for all MD simulations, and the systems were equilibrated in the NVE ensemble for 50,000 steps, followed by equilibration in the NVT ensemble for an additional 50,000 steps. The minimized complexes were used as starting conformations for the MD simulations. Periodic boundary conditions and particle mesh Ewald (PME) electrostatics were employed in the simulations [52]. Finally, 25,000 ps molecular dynamics simulations were performed at 298 K with a trajectory recording interval of 50 ps. In addition, various dynamic analysis parameters, like RMSD, RMSF, Rg, SASA, H-bond, DCCM, and others, were carried out using the GROMACS tool.

### 3.4. Binding Free Energy Calculation

The binding free energy of protein–ligand complexes of all three hits and reference *d*-tubocurarine with nAChRs were calculated using the molecular mechanics generalized Born surface area (MM/GBSA) approach [53]. The following equation was used to calculate the binding free energy from the MD simulation:Δ*G*_bind_ = Δ*E*_MM_ + Δ*G*_solv_(1)
where Δ*G*_bind_ denotes the binding free energy. The changes in the gas phase molecular mechanics (Δ*E*_MM_) and solvation Gibbs energy (Δ*G*_solv_) are determined as follows:Δ*E*_MM_ = Δ*E*_vdw_ + Δ*E*_ele_(2)
Δ*G*_solv_ = Δ*G*_ps_ + Δ*G*_nps_(3)
where Δ*E*_MM_ is the sum of the changes in the Van der Waals energies Δ*E*_vdw_ and the electrostatic energies Δ*E*_ele_. The polar solvation Δ*G*_ps_ was calculated using the generalized Born model, while the non-polar solvation Δ*G*_nps_ was estimated by the solvent-accessible surface area. The solvent dielectric constant of 78.5 and the non-polar surface tension constant of 0.0072 kcal/mol·Å^2^ were used for the MM/GBSA calculations.

### 3.5. ADMET Property Prediction

In this section, the Pre-ADMET server application (https://preadmet.qsarhub.com, accessed on 12 April 2024) was used to calculate ADMET parameters, such as buffer solubility, blood–brain barrier penetration (BBB), plasma protein binding (PPB), cytochrome P450 2D6 inhibition (CYP2D6 inhibition), Ames test, and hERG inhibition. The Pre-ADMET approach is based on different classes of molecular parameters, which are considered for generating quantitative structure properties.

## 4. Conclusions

Through the effective ensemble-based virtual screening approach, including molecular property filters, a 3D pharmacophore model, and molecular docking, three hit compounds were identified as potential NMBAs from the ZINC15 database. In order to further investigate the binding modes between three hits and nAChRs at simulated physiological conditions, the molecular dynamics simulation was performed. Based on the common results of the RMSD, RMSF, Rg, SASA, H-bond, and DCCM, ZINC257459695 displayed stable binding patterns and was considered as a promising lead compound. Furthermore, deeper analysis of the MD results revealed that ZINC257459695 could stably bind to nAChRs’ active site and interact with the key residue Asp165. From the MM-GBSA analysis, the identified ZINC257459695 was the most reliable binding mode for nAChRs, with the binding free energy of −50.40 kcal/mol. Additionally, the ADMET properties revealed that ZINC257459695 can be further developed as potential drug candidates. Overall, ZINC257459695, which possesses huge potential to serve as a promising lead compound in developing novel NMBAs as an adjunct to general anesthesia, warrants further optimization.

## Figures and Tables

**Figure 1 molecules-29-01955-f001:**
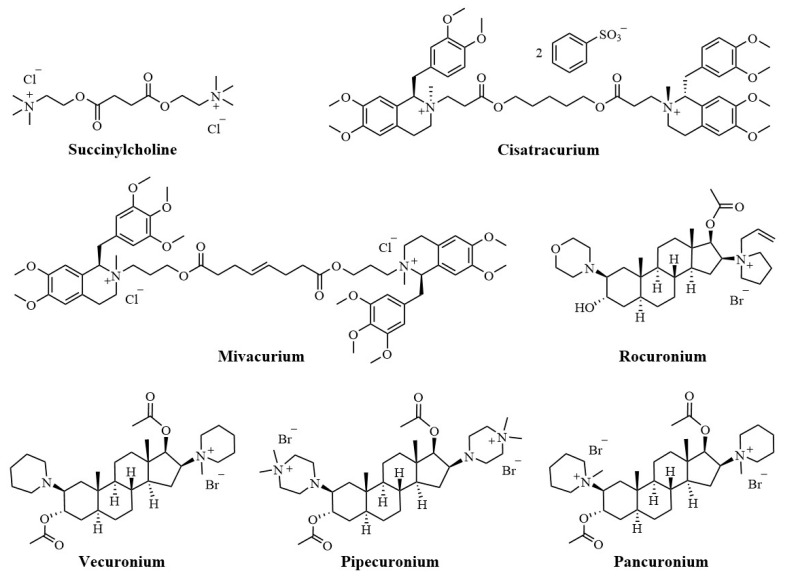
The chemical structures of marketed NMBAs.

**Figure 2 molecules-29-01955-f002:**
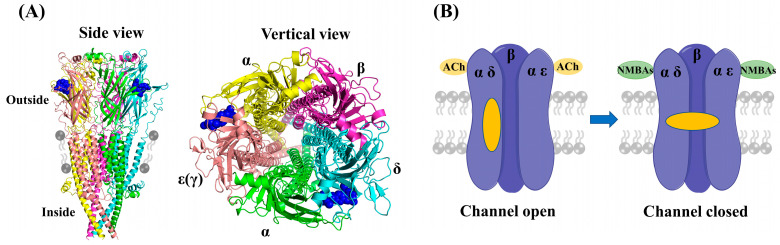
(**A**) The structure of muscular nAChRs (PDB ID: 7SMS). (**B**) The schematic diagram of adult nAChRs inhibited by NMBAs binding both ACh sites.

**Figure 3 molecules-29-01955-f003:**
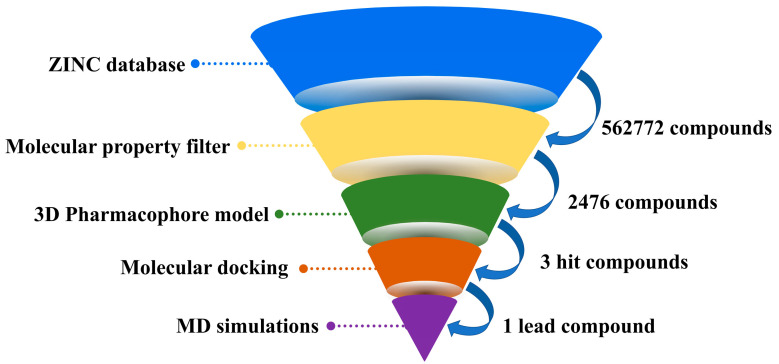
The workflow for discovery of potential NMBAs in this research.

**Figure 4 molecules-29-01955-f004:**
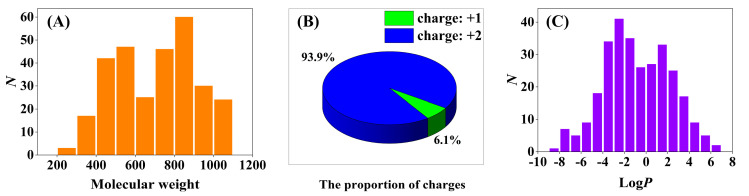
Molecular properties of ligands in the NMBAs database. (**A**) Frequency distribution of molecular weights. (**B**) The proportion of ligands with single-charged versus double-charged. (**C**) Frequency distribution of molecular calculated logP values.

**Figure 5 molecules-29-01955-f005:**
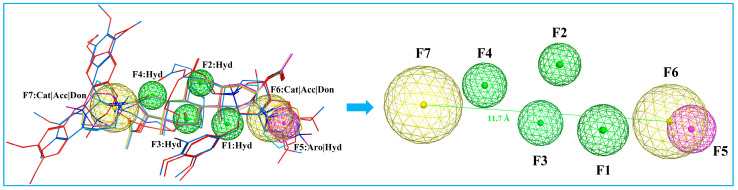
Pharmacophore model based on the aligned NMBAs. Color code: cisatracurium, red; mivacurium, blue; rocuronium, cyan; vecuronium, pink; pipecuronium, magenta; pancuronium, yellow. Pharmacophore features: Hyd, hydrophobic groups; Aro, aromatic rings; Cat, Cation; Acc, H-bond Acceptor; Don, H-bond donor.

**Figure 6 molecules-29-01955-f006:**
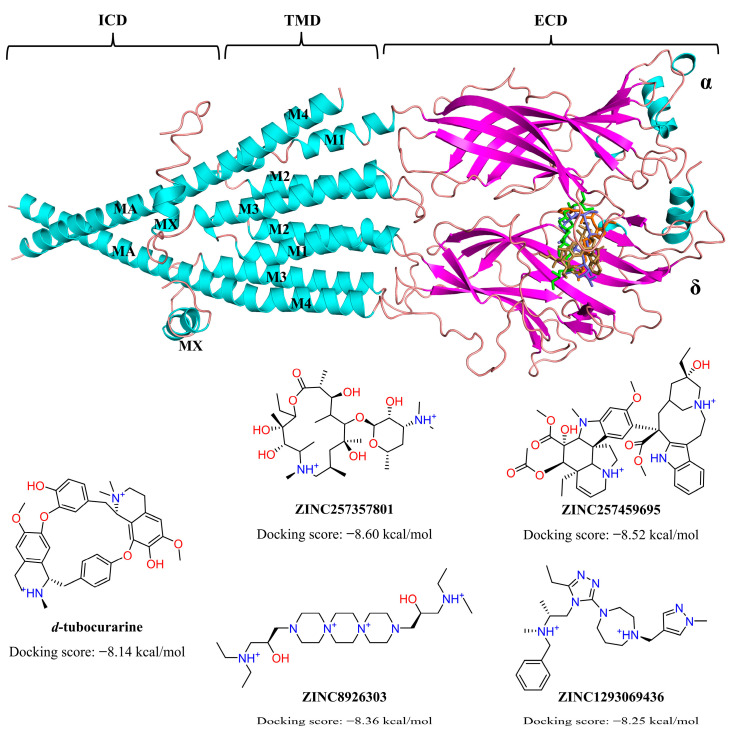
The chemical structures and docking scores (kcal/mol) of screened hit compounds (PDB ID: 7SMS).

**Figure 7 molecules-29-01955-f007:**
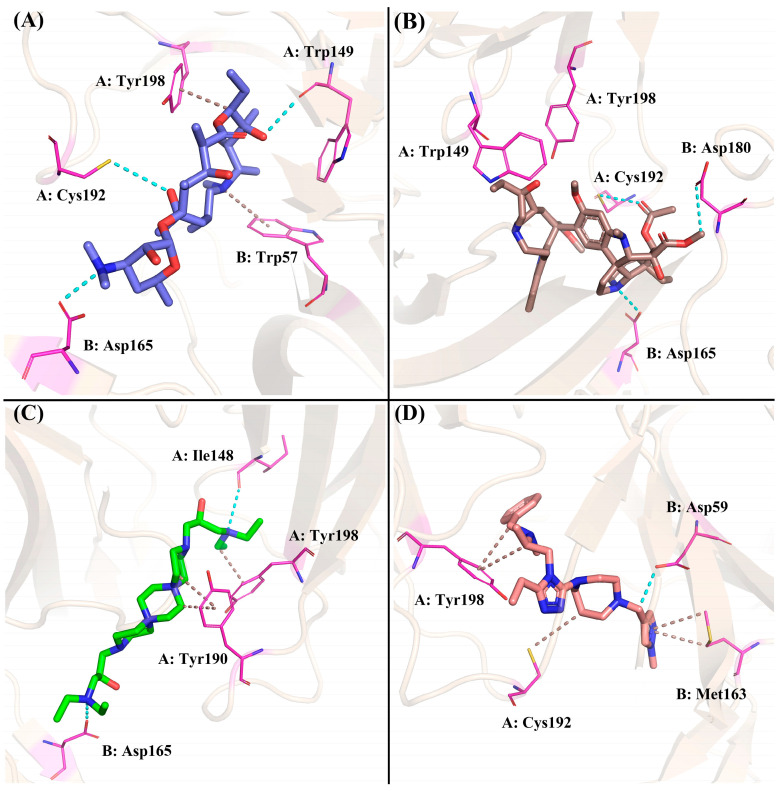
Preferred binding poses of ZINC257357801 (**A**), slate sticks, ZINC257459695 (**B**), sand sticks, ZINC8926303 (**C**), green sticks, and ZINC1293069436 (**D**), salmon sticks, bound to the nAChRs α-δ site (PDB ID: 7SMS).

**Figure 8 molecules-29-01955-f008:**
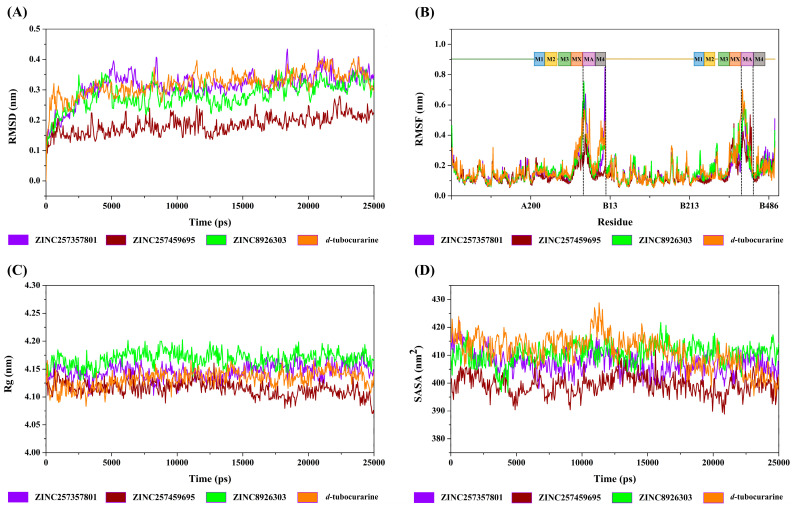
The RMSD (**A**), RMSF (**B**), Rg (**C**), and SASA (**D**) curves of the ZINC257357801, ZINC257459695, ZINC8926303, and *d*-tubocurarine systems.

**Figure 9 molecules-29-01955-f009:**
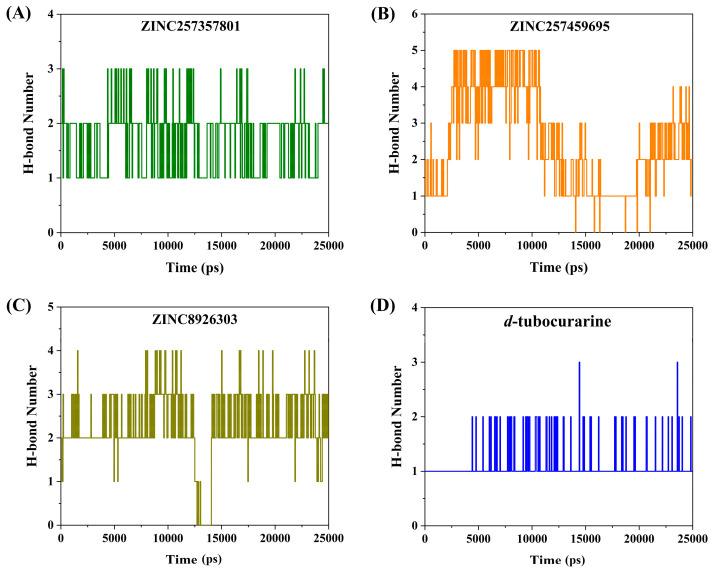
Number of hydrogen bonds for systems: ZINC257357801 (**A**), ZINC257459695 (**B**), ZINC8926303 (**C**), and *d*-tubocurarine (**D**) systems during the MD simulations.

**Figure 10 molecules-29-01955-f010:**
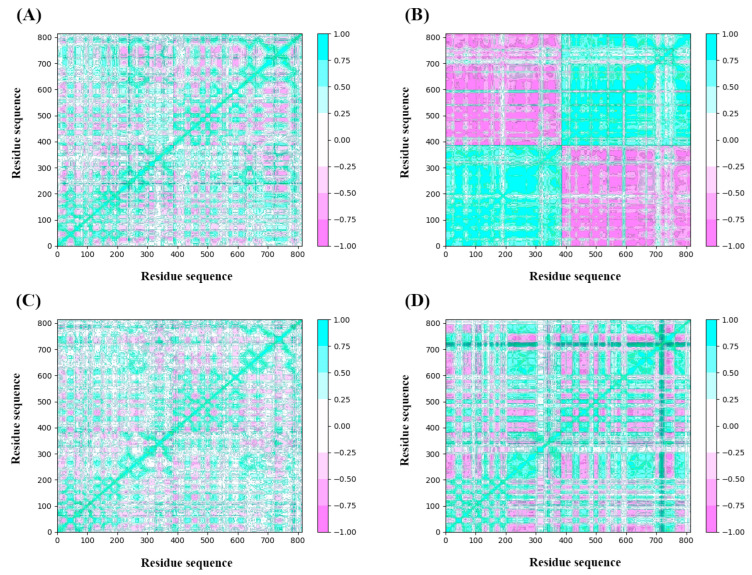
DCCM comparative analysis of ZINC257357801 (**A**), ZINC257459695 (**B**), ZINC8926303 (**C**), and *d*-tubocurarine (**D**).

**Figure 11 molecules-29-01955-f011:**
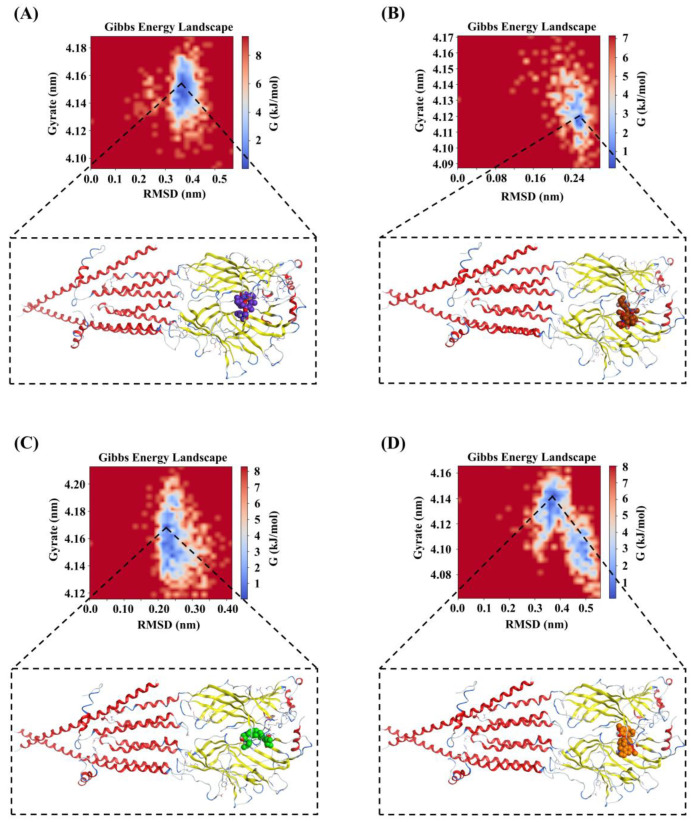
The Gibbs free energy landscapes of ZINC257357801 (**A**), ZINC257459695 (**B**), ZINC8926303 (**C**), and *d*-tubocurarine (**D**) systems during the MD simulations.

**Figure 12 molecules-29-01955-f012:**
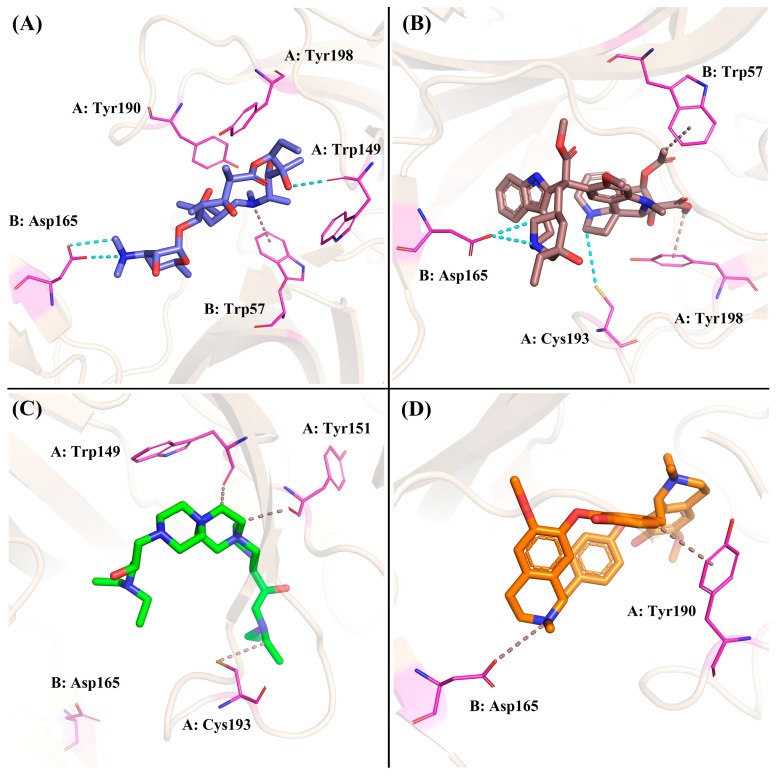
The binding modes of ZINC257357801 (**A**), ZINC257459695 (**B**), ZINC8926303 (**C**), and *d*-tubocurarine (**D**) given by the molecular dynamics simulation.

**Table 1 molecules-29-01955-t001:** Binding free energy calculated by the MM-GBSA method (kcal/mol).

System	Δ*G*_bind_	Δ*E*_vdw_	Δ*E*_ele_	Δ*G*_ps_	Δ*G*_nps_
ZINC257357801	−40.52 ± 5.49	−42.87 ± 3.28	−726.05 ± 4.22	735.10 ± 1.23	−6.70 ± 0.05
ZINC257459695	−50.40 ± 3.61	−33.82 ± 0.84	−1453.27 ± 0.09	1442.28 ± 3.51	−5.59 ± 0.00
ZINC8926303	−31.01 ± 6.22	−37.21 ± 2.20	−586.86 ± 3.01	598.35 ± 4.97	−5.28 ± 0.02
*d*-tubocurarine	−39.57 ± 3.11	−55.84 ± 0.11	−657.70 ± 2.99	680.80 ± 0.83	−6.82 ± 0.16
Cisatracurium	−66.86 ± 3.07	−92.50 ± 2.01	−669.38 ± 0.33	706.26 ± 2.30	−11.24 ± 0.08
Rocuronium	−57.65 ± 2.50	−62.59 ± 1.72	−340.73 ± 1.67	353.45 ± 0.70	−7.78 ± 0.01

**Table 2 molecules-29-01955-t002:** The ADMET prediction for the investigated compounds.

Compound	Buffer Solubility ^1^	BBB ^2^	PPB ^3^	CYP2D6 Inhibition	Ames Test	hERG Inhibition
ZINC257357801	1132	0.04	3.83	Inhibitor	Non-mutagen	Ambiguous
ZINC257459695	0.30	0.11	13.97	Non	Non-mutagen	High
ZINC8926303	88,380	0.05	19.16	Inhibitor	Mutagen	Ambiguous
*d*-tubocurarine	1.18	1.22	67.68	Non	Non-mutagen	High risk
Cisatracurium	0.0081	0.80	72.49	Non	Non-mutagen	Medium risk
Rocuronium	121.69	0.26	18.97	Inhibitor	Mutagen	Low risk

^1^ Buffer solubility: water solubility in buffer system (SK atomic types, mg/L). ^2^ BBB: blood–brain barrier penetration (C.brain/C.blood). ^3^ PPB: plasma protein binding (%).

## Data Availability

The raw data supporting the conclusions of this article will be made available by the authors, without undue reservation.

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
