# Peer review of "Ensemble-Based Virtual Screening Led to the Discovery of Novel Lead Molecules as Potential NMBAs"

_molecules, 2024, doi:10.3390/molecules29091955_

Round 1

Reviewer 1 Report

Comments and Suggestions for Authors

The authors have devised and implemented a screening protocol to uncover potential neuromuscular blocking agents beyond those already on the market. The paper is well-structured, containing detailed information and clear, explanatory figures. The methods applied, primarily utilizing commercial software, appear robust, mirroring the credibility of the results. Overall, the paper merits consideration for publication in the journal. However, prior to publication, the authors should review the manuscript for English grammar, particularly to address several instances where main verbs are missing in sentences.

To the best of the reviewer's knowledge, the applied protocol is new and not completely Implemented in any commercial code. The protocol which follows 4 steps: Molecular Property Filtering -> Pharmacophore Modelling -> Molecular docking -> MD,  beside isolation allows to disclose the interaction mechanisms between isolated NBA and the Muscular nAChRs as well.

Thanks to the applied protocol the authors were able to isolate a potential NBA starting from a database of about 1000 million compounds. Only synthesis could provide information on the toxicity or efficacy of the new compound.  

The Pharmacophore Model relies on a set of seven descriptors. It could be interesting to show and analyze the effect of modifying types and the number of descriptors on the isolated compounds.

The authors could compare the bindings free energies of some of the marketed NBA compounds with those of Table 1. The comparison would provide further evidence of the potential efficacy of the proposed NBA.

References are appropriate.

Figures are clear and explanatory. English grammar needs to be carefully checked.

Comments on the Quality of English Language

Prior to publication, the authors should review the manuscript for English grammar, particularly to address several instances where main verbs are missing in sentences.

Author Response

File attached!

Reviewer 2 Report

Comments and Suggestions for Authors

File attached!

Author Response

File attached!

Round 2

Reviewer 2 Report

Comments and Suggestions for Authors

After add the suggested data, I suggest accepting the manuscript in this journal.